# Do University Students Adhere to WHO Guidelines on Proper Use of Face Masks during the COVID-19 Pandemic?—Analysis and Comparison of Medical and Non-Medical Students

**Marta Szepietowska** [1], **Piotr K. Krajewski** [2], **Łukasz Matusiak** [2], **Rafał Białynicki-Birula** [2] and **Jacek C. Szepietowski** [2,*]

[1] Student Research Group of Experimental Dermatology, Department of Dermatology, Venereology and Allergology, Wroclaw Medical University, 50-368 Wroclaw, Poland; marta.szepietowska0703@gmail.com

[2] Department of Dermatology, Venereology and Allergology, Wroclaw Medical University, 50-368 Wroclaw, Poland; piotr.krajewski@student.umed.wroc.pl (P.K.K.); luke71@interia.pl (Ł.M.); rafal.bialynicki-birula@umed.wroc.pl (R.B.-B.)

[*] Correspondence: jacek.szepietowski@umed.wroc.pl; Tel.: +48-71-784-22-86

**Abstract:** Covering the mouth and nose has become the common standard for protection from human-to-human virus transmission during the COVID-19 pandemic. The aim of the study was to investigate whether students at universities (medical and non-medical students) do adhere to WHO recommendations on the proper use of face protection. The study was based on the survey created with Google® Forms regarding data on demographics and self-reported skin conditions. The main questions addressed the WHO guideline on the correct use of face masks. Responses were obtained from 1173 students. Compliance with all WHO criteria among the whole group of respondents was very low at less than 1% with no significant difference between medical and non-medical students. Covering of the nose and mouth with the face mask was the criterion most commonly complied with (81.2%); contact avoidance of touching the mask with hands appeared to be the most difficult criterion to comply with (2.8%). Medical students were significantly more compliant with three out of seven criteria. In general, compliance with the WHO guidelines on the proper use of face masks was dramatically low among all the students. Education campaigns should be introduced to increase the awareness of the correct use of face coverings during the viral pandemic.

**Keywords:** COVID-19; face masks; WHO guidelines; SARS-CoV-2; university students; medical students; non-medical students

## 1. Introduction

Covering the mouth and nose has become the common standard in protection from human-to-human virus transmission during the Coronavirus Disease 2019 (COVID-19) pandemic [1,2]. In many countries, wearing face masks has become obligatory for being in a public space. In the first months of Severe Acute Respiratory Syndrome Coronavirus 2 (SARS-Cov-2) spreading, we documented some inconveniences of face-mask use, such as itching, sweating, difficulty in breathing or misting up of glasses, which could negatively influence their protective role [3]. In the, meantime several organizations, including the World Health Organization (WHO), announced their guidelines concerning the proper use of face masks [4]. Adherence to those recommendations seems to be of crucial importance, as improperly used face masks may not offer the expected protection and even could predispose an individual to the promotion of virus transmission [2,4].

The aim of this study was to evaluate whether students at universities adhere to international recommendations on the correct use of face masks. Additionally, we were interested to see whether there are differences between medical and non-medical students in their attitudes and practices concerning proper mouth and nose covering.

## 2. Materials and Methods

The study was based on the survey created with Google® Forms and posted on numerous Facebook® groups for students in Poland. We used a snowball sampling technique implemented by Heckathorn et al. [5] in which every responder could resend the link to the survey and invite further participants. This way, we collected a representative sample of 1173 students (confidence level, 95%; margin of error, 2.86%). The study was performed based on the statutory activity of the department in accordance with the ethical approval of the Wroclaw Medical University Institutional Review Board (ST.C260.18.019). The questionnaire included some demographics, self-reported sensitive skin, atopic predisposition, and the presence of facial dermatosis and face mask-induced itch. The crucial part of the questionnaire comprised seven questions of WHO guidance on the correct use of face masks (Table 1).

**Table 1.** Questions used in the survey based on WHO guideline criteria concerning the correct use of face masks.

| Criterion | Question |
|---|---|
| C1 | How often does the mask strictly cover your mouth and nose at work? |
| C2 | How often do you happen to touch the mask at work? |
| C3 | How often do you take the mask off properly at work (without touching its anterior surface)? |
| C4 | How often do you clean/disinfect your hands after taking off/touching the mask at work? |
| C5 | How often do you replace the mask when it dampens at work? |
| C6 | How often do you reuse single-use masks? |
| C7 | How often do you dispose of single-use masks after one use at work? |

The respondents were asked to answer each question as "always/nearly always", "often", "rarely" or "never". Compliance with a criterion was documented when the chosen answer was "always/nearly always". We decided to omit the question on hand hygiene and disinfection procedures prior to putting on the face mask (originally included in the WHO guidelines), as at that time, the Polish government introduced, again, mandatory face coverings for being in a public space. All students wore face masks upon entering the universities. Therefore, having this question in our survey might have caused bias in our final results. The used methodology has been implemented in medical studies for a long time [6]. Due to the abovementioned snowball effect, the online survey spread quickly among the interested participants. Until now, hundreds of articles in worldwide journals have been published using this methodology [7–11]. Moreover, with the COVID-19 pandemic, researchers have been using online questionnaires even more frequently due to contact restrictions, lack of face-to-face contact and no possibility to travel [12–18]. A similar methodology was employed by Machida et al. [19] in their paper about the incorrect use of face masks during the COVID-19 pandemic. Although easy, cheap and efficient, it is important to emphasize that the online survey technique has its limitations. Firstly, the researchers are unable to assess the response rate of the questionnaire. Moreover, as in our study, all the data are self-reported, which may be problematic in the assessment of the prevalence of skin conditions. The data were collected between the 1st and 7th of October 2020.

The statistical analysis was performed with the Chi-square test and logistic regression analysis where appropriate, using Statistica 13.3 (TIBCO Software Inc. Palo Alto, CA, USA) software. The *p*-values less than 0.05 were considered significant.

## 3. Results

Responses were obtained from 1173 students (876 females–74.7% and 297 males–25.3%). The mean age of the group was $20.9 \pm 2.9$ years (age range: 17–27 years). Among the whole group, there were 665 medical students (56.7%) and 508 (43.3%) students of

other faculties not related to health sciences (non-medical students). More than half of the respondents declared sensitive skin and the presence of facial dermatosis (52.3% and 57.5%, respectively). Atopic predisposition was reported by 33.8%, and 20.2% experienced face mask-induced itch. There was no significant difference in the prevalence of the abovementioned signs and symptoms between medical and non-medical students (Table 2).

**Table 2.** Group characteristics.

| Characteristics | | All Patients (*n* = 1173) |
| --- | --- | --- |
| Sex | Men | 297 (25.3%) |
| | Women | 876 (74.7%) |
| Age (years) | Mean | 21 |
| | Standard deviation | 2.9 |
| University | Medical | 665 (56.7%) |
| | Non-medical | 508 (43.3%) |
| Sensitive skin | All | 613 (52.3%) |
| | Medical | 335 (50.4%) |
| | Non-medical | 278 (54.72%) |
| Facial dermatosis | All | 674 (57.5%) |
| | Medical | 369 (55.5%) |
| | Non-medical | 305 (60%) |
| Atopic predisposition | All | 396 (33.8%) |
| | Medical | 221 (33.2%) |
| | Non-medical | 175 (34.5%) |

Among the whole group of respondents, only three students (0.3%) were compliant with all WHO criteria on the correct use of face mask. There was no significant difference between medical and non-medical students (0.2% and 0.4%, respectively) (Table 3).

**Table 3.** Adherence to WHO guideline criteria concerning the correct use of face masks among students.

| Criterion | All Students (*n* = 1173) | Medical Students (*n* = 665) | Non-Medical Students (*n* = 508) | *p* |
| --- | --- | --- | --- | --- |
| All criteria | 3 (0.3%) | 1 (0.2%) | 2 (0.4%) | 0.71590 |
| C1 | 952 (81.2%) | 570 (85.7%) | 382 (75.2%) | **0.00001** |
| C2 | 33 (2.8%) | 19 (2.9%) | 14 (2.8%) | 0.91725 |
| C3 | 394 (33.6%) | 259 (39.0%) | 135 (26.6%) | **0.00001** |
| C4 | 263 (22.4%) | 150 (22.6%) | 113 (22.2%) | 0.89888 |
| C5 | 330 (28.1%) | 203 (30.5%) | 127 (25.0%) | **0.03700** |
| C6 | 288 (24.6%) | 126 (19.0%) | 162 (31.9%) | **0.00000** |
| C7 | 309 (26.3%) | 158 (23.8%) | 151 (29.7%) | **0.02156** |

N—number of participants, bold—*p* < 0.05.

Upon analyzing particular criteria, strict covering of the nose and mouth with a face mask (C1) was most commonly complied with (81.2%). On the other hand, avoidance of touching the mask with hands (C2) appeared to be the most difficult criterion to comply with (2.8%). We were unable to disclose any significant differences in compliance with criteria between females and males (detailed data not showed).

Medical students, in comparison to non-medical ones, showed significantly better compliance with C1 (strict covering of the nose and mouth with the face mask; 85.7% and 75.2%, respectively), C3 (taking off the mask properly without touching the anterior surface; 39.0% and 26.6%, respectively) and C5 (replacing the mask when it dampens; 30.5% and 25.0%, respectively). In contrast, significantly more non-medical students adhered to the C6 criterion (not reusing single-use masks; 31.9% and 19.0%, respectively). Non-medical students also complied statistically better with the last WHO criterion, C7 (disposure of single-use masks after their use at work; 29.7% and 23.8%, respectively) (Table 2).

Self-reported sensitive skin and face mask-induced itch predisposed respondents to the lack of adherence to the C1 criterion (strict covering of the nose and mouth with the face mask; odds ratio (OR) 0.7, $p = 0.001$ and OR 0.58, $p = 0.0007$, respectively). Interestingly, students with sensitive skin and facial dermatosis were more prone to comply with C5 (replace the mask when it dampens; OR 2.65, $p < 0.0001$ and OR 1.63, $p = 0.0002$). Moreover, those with sensitive skin and atopic predisposition appeared to be more compliant with C4 (washing/disinfecting hands after face mask removal or touching; OD 1.39, $p = 0.01$ and OD 1.44, $p = 0.006$, respectively).

## 4. Discussion

Education during the COVID-19 pandemic appeared to be a special challenge for both students and teachers. In many countries, the education program was moved, wholly or partially, to the virtual world. In Poland, after the summer holiday, all students returned to their universities, starting face-to-face classes. Gathering a bigger number of people in one place created an increased risk of SARS-CoV-2 infections. Therefore, they were all obliged to wear face masks.

To the best of our knowledge, our study is the first one investigating student compliance with guidelines on the correct use of face protection. Unfortunately, our results showed that adherence to all WHO criteria among the studied students was very low, with less than 1%. In a similar study carried out on the Japanese general population, Machida et al. [19] reported that 23.1% adhered to those criteria. However, it is important to notice that we employed more strict rules for fulfilling the criteria, accepting only answers of "always/nearly always". In the Japanese study, the answer "sometimes" also qualified the respondents as compliant with WHO criteria [19]. This can at least partly explain the observed difference between the results of both studies. Adherence to the face mask proper usage guidelines is not an unknown problem, even among health care workers (HCW). In 2019, it was shown that only 18% of HCW involved in surgical procedures fully adhered to the Centers For Disease Control (CDC) recommendations [20].

We also noted significant relationships between self-reported skin complaints and adherence to some of the WHO guideline criteria. It is not surprising that those respondents who declared sensitive skin and face mask-induced itch were less compliant with strict covering of the nose and mouth. Sensitive skin is also an itchy condition [21]. Face mask wearing with strict attachment to the skin surface creates a specific occlusion that may contribute to an increased itching sensation [22]. Interestingly, those with the presence of facial skin disease were more prone to adhere to the criterion for replacing the mask when it dampens. This could at least partly be explained by the fact that wearing face masks was shown to exacerbate such conditions as acne, rosacea, seborrheic dermatosis [23,24], and the respondents tried to avoid additional flare-ups caused by a wet, occlusive environment.

Although the adherence to particular criteria was low, medical students showed to be more compliant with some criteria, such as strict covering of the nose and mouth with the face mask, taking off the mask properly without touching the anterior surface and

replacing the mask when it dampens. Previously, during the beginning of the COVID-19 pandemic when face mask use was not mandatory and only recommended, we revealed that medical students wore face protection significantly more often than non-medical students. Moreover, medical students more commonly used single-use masks and wore face protection for longer periods of time [25]. This is not surprising, as the differences in health-related attitudes and behaviors between medical and non-medical students have been previously documented [26–28]. The criterion that non-medical students complied significantly better with was the disposure of single-use masks after their use at work. We believe that it may be caused by differences in types of university classes. Clinical practice in hospitals and the students' attendance at different wards on the same day would require the use of several masks daily. In comparison to other university classes, it is both physically and financially more difficult.

We are aware of some limitations of our project. Using the employed methodology, we were unable to assess the response rate. However, such a methodology has been previously successfully used by several authors [2,29,30]. Using this questionnaire, we gathered information from a sample that could be representative of the Polish student population in one week (CI 95%, margin of error 2.86%). Nevertheless, our cohort does not fully reflect the Polish student group. The female predominance in our study might be attributed to the greater interest and willingness of women to take part in questionnaire studies, as well as the higher proportion of female medical students. Moreover, the study was based on self-reported adherence and self-reported assessment of skin conditions. More objective methods with external observers and skin examinations by physicians might add value to future studies. Furthermore, our study was based only on the Polish population. Ideally, it should include students from different geographic regions, as well as different cultures. This makes it impossible to extrapolate our results on the worldwide student population.

## 5. Conclusions

In conclusion, the results of the current study confirm that, most probably, the awareness, knowledge and personal interests of medical students may contribute to their more correct usage of face protection against SARS-CoV-2 transmission. In spite of this, in general, compliance with the guidelines on the proper use of face masks is dramatically low among all the students. Although we are far from extrapolating our results to the general population, due to the female predominance and concentration on a student population, it seems that special education campaigns should be introduced to increase the awareness of the correct use of face coverings during viral pandemics. It is important to understand that, even among people who wear masks frequently, its incorrect use may decrease its ability to protect and therefore increase the spread of SARS-CoV-2. Lastly, in the era of vaccination and the possible eradication of SARS-CoV-2, it is still unclear whether another pandemic will occur in the nearest future. Hence, as medical professionals, we should try our best to educate people on the importance of correct face mask use in the prevention of respiratory infections.

**Author Contributions:** Conceptualization, P.K.K., Ł.M., R.B.-B., M.S. and J.C.S.; methodology, Ł.M., R.B.-B. and J.C.S.; formal analysis P.K.K., Ł.M. and J.C.S.; investigation, P.K.K., Ł.M., R.B.-B., M.S. and J.C.S.; data curation, P.K.K., Ł.M. and J.C.S.; writing—original draft preparation, P.K.K. and M.S.; writing—review and editing, Ł.M., R.B.-B. and J.C.S.; visualization, P.K.K. and M.S.; supervision, Ł.M., R.B.-B. and J.C.S.; All authors have read and agreed to the published version of the manuscript.

**Funding:** This research received no external funding.

**Institutional Review Board Statement:** The study was conducted according to the guidelines of the Declaration of Helsinki, and approved by the Wroclaw Medical University Institutional Review Board (ST.C260.18.019).

**Informed Consent Statement:** Informed consent was obtained from all subjects involved in the study. The study was performed based on the statutory activity of the department, in accordance with the ethical approval of the Wroclaw Medical University Institutional Review Board (ST.C260.18.019).

**Data Availability Statement:** Data available on request.

**Conflicts of Interest:** The authors declare no conflict of interest.

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
