# Peer review of "Do University Students Adhere to WHO Guidelines on Proper Use of Face Masks during the COVID-19 Pandemic?—Analysis and Comparison of Medical and Non-Medical Students"

_applsci, doi:10.3390/app11104536_

Round 1

Reviewer 1 Report

Thank you for the opportunity to review this article, it is an interesting topic that is worth to be analyzed.
There is an issue that requires further review, I have seen nowhere in the paper the approval by the local research board of the study protocol, it is a basic rule, without approval of the ethics committee can not do this type of research. Please clarify this aspect in an unequivocal way.

A table with the overall description of the cohort and by health sciences and non-health sciences students should be included.

Throughout the paper they confuse COVID-19 disease with SARS-CoV-2, please review and clarify terms.

Author Response

Ad. Reviewer 1:

Thank you very much for the review and for your time. We have revised our manuscript according to your comments. All the introduced changes are highlighted in the manuscript.

  1. Thank you for the opportunity to review this article, it is an interesting topic that is worth to be analyzed.

  • Thank you very much for this comment. We are pleased that you found our paper of value. We do believe that the topic of face masks use should be addressed more and more frequently.

  1. There is an issue that requires further review, I have seen nowhere in the paper the approval by the local research board of the study protocol, it is a basic rule, without approval of the ethics committee can not do this type of research. Please clarify this aspect in an unequivocal way.

  • We acknowledge your comment. We have added it in the Methods section: “The study was performed based on the statutory activity of the department, in accordance to the ethical approval of the Wroclaw Medical University Institutional Review Board (ST.C260.18.019)”.

  1. A table with the overall description of the cohort and by health sciences and non-health sciences students should be included.

  • Thank you for this comment. We have added the table comparing both groups.

  1. Throughout the paper they confuse COVID-19 disease with SARS-CoV-2, please review and clarify terms.

  • Thank you very much for this comment. We have analyzed our use of COVID – 19 and SARS-CoV-2 throughout the paper and corrected possible confusions. We have additionally explained the acronyms at the beginning of the paper

Reviewer 2 Report

the file is attached

Author Response

Ad. Reviewer 2:

We appreciate Reviewer’s careful revision of our manuscript, which help us to increase the quality of our paper. The manuscript was corrected and modified:

  1. The article is relevant and analyses an important issue facing the current pandemic situation. A clear and concise language is used to present the study. The most relevant weakness of this manuscript is the absence of discussions about the methodology used in this article.

  • Thank you very much for this comment and for agreeing that it is important issue. We have analyzed our paper and corrected it according to your suggestions. We believe that this will improve the article and that it will be suitable for publication.

  1. Title: Please improve the title and include key words (e.g. university students) to make it understandable. This article is to explore if students at universities (medical and non-medical students) do adhere to WHO recommendations on the proper use of face protection. The title must be representative of the article contents.

  • We acknowledge your comment. We have rewrite the title to “Do university students adhere to the WHO guideline on proper use of the face masks during COVID-19 pandemic? – analysis and comparison of medical and non-medical students” and added three key words: university students; medical students; non-medical students

  1. This manuscript only includes 16 references. Research papers must be grounded on theoretical background, including in depth literature review discussion, and therefore much more references.

  • Thank you for your remark. We have revised our paper according to your instructions, which markedly increased the number of references. No we have 30 references in our manuscript.

  1. I miss the literature review about the methodology regarding survey in this article. I recommend to the authors to add some relative literatures about the issues of online survey.

  • Thank you for this comment. Indeed, the used methodology is not a standard one, however practiced a lot in medical research. We have included the short description, advantages and disadvantages of this technique: “The used methodology has been implemented in medical studies for a long time [6]. Due to the above mentioned snowball effect, the online survey spread quickly among the interested participants. Until now, hundreds of articles in worldwide journals have been published using this methodology [7-11]. Moreover, with the COVID-19 pandemic, researchers have been using online questionnaires even more frequently due contact restrictions, lack of face to face contact and possibility to travel [12-18]. Similar methodology was employed by Machida et al. [19] in their paper about the incorrect use of face masks during COVID-19 pandemic. Although easy, cheap and efficient, it is important to emphasize that online survey technique has its limitation. Firstly the researchers are unable to assess response rate of the questionnaire. Moreover, as in our study, all the data is self-reported, which may be problematic in assessment of prevalence of skin conditions”

  1. Research findings are of scientific value only if they can be generalized. At the very least, it should be possible to generalize from the sample to the population from which the sample was drawn. This can happen only if the sample is representative of the population, which requires an important condition to be fulfilled. It is that a valid method of sampling should have been adopted; a method that recruits a sample that is overrepresented for a particular characteristic cannot represent the population. The authors should explain the reasons why your sampling method with Google Forms is appropriate, and if 876 females out of 1173 students (74.7%) is representative of the population.

  • Thank you for this comment. According to the available reports of Central Statistical Office of Poland, in 2018/2019 there was 893 thousand students studying at public universities. Our group of 1173 gives us a confidence level of 95% with exactly 2.86% of margin of error. This means that the conclusions drawn from this paper could be generalized on the population of Polish students. Moreover, it is true that more women study at Polish Universities (56.9% vs 43.1%). Even bigger predominance of female students in our cohort is caused by the predominance of women studying at medical universities and greater willingness of taking part in survey – based studies.

  1. Please discuss more your research limitations.

  • The limitations were described more thoroughly. The following statement was added “Using this questionnaire we gathered information from a sample that could be representative of Polish students population in one week (CI 95%, margin of error 2.86%). Nevertheless, our cohort does not fully reflect Polish students group. The female pre-dominance in our study might be attributed to greater interest and willingness of women to take part in questionnaire studies, as well as higher proportion of female medical students” and “Furthermore, our study was based only on Polish population. Ideally, it should include students from different geographic regions, as well as different cultures. This makes impossible to extrapolate our results on worldwide student population.”

  1. Conclusions: I think they should be strengthened, include the limitations of the article and the potential for future research.

  • Thank you for your comment. We have added the following statement to the conclusions: “It is important to understand that even among people who wear masks frequently, its incorrect use may decrease its ability to protect and therefore increase spread of SARS-CoV-2. Lastly, in the era of vaccination and possible eradication of SARS-CoV-2, it is still unclear whether another pandemic occurs in the nearest future. Hence, as medical professionals, we should try our best and educate people on the importance of correct face mask use in prevention of respiratory infections”

  1. Page 2, line 63, please confirm if “was 20.9 2.9 years…” is correct or just a typo.

  • Thank you for this remark. There was supposed to be a “±” symbol. It was added

  1. Page 3, line 72, please check if “2.0%” is correct. It seems to me that the percentage should be 0.15%, and in Table 2 as well. Furthermore, the p-value for All criteria is 0.71590 which means there was no significant difference between medical and non-medical students. However, line 71 showed “with significant difference…”. Please confirm your data and statement.

  • Thank you for this comment. You are completely right. There was a typo in the percentage in the table, therefore it later appeared in the text. Regarding the difference, following your advice, we have added “with NOT significant difference”

  1. Page 4, line 92, there is a typo “p=0002", please correct it.

  • Thank you for this remark. It was corrected.

  1. The p-value for C7 is 0.02156 which means there was significant difference between medical and non-medical students. However, the authors did not indicate and explain this result in Page 3.

  • Thank you for this comment. We have addressed this in Discussion Section. The following statements were added: “The only criterion which non-medical students complied significantly better was the disposure of single-use masks after their use at work. We believe that it may be caused by differences in types of university classes. The clinical practice in hospitals and students’ attendance at different wards on the same day would require use of several masks daily. In comparison to other university classes, it was both physically and financially more difficult”. Moreover, we added to the Results:  “Non-medical students also complied statistically better with the last WHO criterion, C7 (disposure of single-use masks after their use at work (29.7% and 23.8%, respectively)”

  1. A greater effort is required to identify the contributions derived from this study. This is a prominent issue where authors should pay more attention to address it.

  • Thank you for you remark. We believe that the change of wording and statements added according to your suggestions have improved the contributions of this study.

Reviewer 3 Report

Dear Authors, I appreciated your simple but significant contribute in this hystorical COVID age. Introduction is actual, methodology is clear.In my opinion your work is acceptable for the publication. I think that it could be considered as a perspective rather than an article.

Author Response

Ad. Reviewer 3:

Thank you very much for this kind review and for your time. We are grateful that you consider this paper acceptable for publication.

Round 2

Reviewer 2 Report

Dear Authors:

Thank you for taking my comments into consideration.  All my previous comments were included. However, I still found a typo. In Table 2, the standard of deviation of Age (years) is 2.9, not "2,9".  Thus, I recommend publishing this article, after correcting the typo and meeting the publication standards of Applied Sciences.

Best Regards,